# Energy Consumption Evaluation of a Routing Protocol for Low-Power and Lossy Networks in Mesh Scenarios for Precision Agriculture

**DOI:** 10.3390/s20143814

**Published:** 2020-07-08

**Authors:** Frederico O. Sales, Yelco Marante, Alex B. Vieira, Edelberto Franco Silva

**Affiliations:** Graduate Program in Computer Science—UFJF, Juiz de Fora-MG 36036-330, Brazil; frederico.sales@ice.ufjf.br (F.O.S.); yelco.marante@ice.ufjf.br (Y.M.); alex.borges@ice.ufjf.br (A.B.V.)

**Keywords:** internet of things, energy consumption, routing protocol for low-power and lossy networks, precision agriculture, mesh networks

## Abstract

Sensor nodes are small, low-cost electronic devices that can self-organize into low-power networks and are susceptible to data packet loss, having computational and energy limitations. These devices expand the possibilities in many areas, like agriculture and urban spaces. In this work, we consider an IoT environment for monitoring a coffee plantation in precision agriculture. We investigate the energy consumption under low-power and lossy networks considering three different network topologies and an Internet Engineering Task Force (IETF) standardized Low-power and Lossy Network (LLN) routing protocol, the Routing Protocol for LLNs (RPL). For RPL, each secondary node selects a better parent according to some Objective Functions (OFs). We conducted simulations using Contiki Cooja 3.0, where we considered the Expected Transmission Count (ETX) and hop-count metric (HOP) metrics to evaluate energy consumption for three distinct topologies: tree, circular, and grid. The simulation results show that the circular topology had the best (lowest) energy consumption, being 15% better than the grid topology and 30% against the tree topology. The results help the need to improve the evolution of RPL metrics and motivate the network management of the topology.

## 1. Introduction

Wireless Sensor Networks (WSNs) are composed of low-cost nodes with limited processing, communication, and detection capabilities, which interact cooperatively to perform complex monitoring tasks in a geographic area of interest [1]. Current and future applications are comprised of many important domains, such as smart cities, environmental monitoring, distributed sensing in industrial plants and health care, as well as the various areas of the Internet of Things (IoT) [2,3].

WSNs present challenging issues related to the reliability of communication and the efficient use of the node’s battery [4]. In particular, routing is an essential problem in these networks due to possible communication failures, limited bandwidth, and energy/power available [5]. Thus, sophisticated techniques are needed to configure and maintain reliable paths, as well as to detect link failures promptly, without wasting energy and communication resources [6].

In this work, we observe the benefits of previously known specific topology characteristics to conduct the network setup where limited resources can compromise all monitored environments. To demonstrate the practical benefits of this research, a real environment was considered. Thus, we focused on Precision Agriculture (PA), more specifically a real coffee plantation in Southeast Brazil. Figure 1 shows the three different network topologies that will be investigated in this work. Similarly, the benefits of knowing the virtual topology can help the Low-power and Lossy Network (LLN) routing protocols obtain the maximum lifetime in terms of energy consumption. In this way, our study was conducted on the Contiki Cooja platform, an open, portable, and multitasking platform to simulate sensor networks. The platform was developed for sensors with low computational and energy resources, where the typical sensor has 2 KB of RAM and 40 KB of ROM. Our main contribution is identifying the best topology and characteristics to meet the goal of balancing low cost, reliability, and the longest useful battery lifetime in a mesh WSN for the IoT environment.

This work focuses on the benefits of fitting the network topology for PA scenarios using a mesh and Internet Engineering Task Force (IETF) standardized LLN protocol with both of its well-known objective functions, comparing them in relation to energy consumption. The results show that the Routing Protocol for LLNs (RPL) has an excellent performance in terms of the sensor network lifetime. The algorithm never leaves a node orphaned (when a parent is no longer eligible, the protocol seeks the best immediate substitute to fill this gap, preventing the isolation of any node in the network). Another good result is how significant the influence of the topology on energy consumption is. Our main results showed that the circular topology proved to be the best in terms of energy consumption compared to the grid and tree topologies, obtaining energy-savings of 30% and 15%, respectively.

The remainder of this paper is organized as follows: in Section 2, we present the background of WSNs. Section 3 presents the related work. Section 4 introduces the evaluation methodology and numerical results. Finally, Section 5 concludes this works and presents future works.

## 2. Background

### 2.1. Wireless Sensor Networks

A WSN is a wireless network for distributed devices, using sensors to monitor cooperatively the physical or environmental conditions, such as temperature, sound, vibration, pressure, movement, etc. Recent advances in wireless communication and electronics have enabled the development of low-power, low-cost multifunctional sensor nodes. These sensors can detect, process, and send/receive information over short distances. In general, they are present in a dense form, to collect high precision data and perform other complex tasks related to both the collection and dissemination of information.

One of the sensor nodes’ main characteristics is their energy limitation, computational resource limitation, and access to the network. However, it is necessary for WSNs to be scalable, robust to failure, self-organizing, and to have a long service life across the network [1]. Therefore, the protocols and the application algorithm involved must be light, resulting in a longer network lifetime without compromising its performance.

### 2.2. RPL

The IETF Routing Over Low-power and Lossy networks (ROLL) working group standardized RPL, which is a gradient-based routing protocol for WSNs with bidirectional links. It can support a wide variety of different link layers, including those that are restricted, potentially lossy, or typically used in conjunction with host devices or routers with very limited resources [7].

RPL is a lightweight proactive routing protocol and requires only small power and memory resources to operate, making it an excellent choice for networks with limitations, such as those of the IoT. RPL is flexible and customizable to adapt to the requirements of heterogeneous IoT applications. This is possible with the help of Objective Functions (OF). The OFs are based on metrics to indicate the best existing routes [8].

Routing metrics are selected according to the network operation or application requirements. They try to maximize or, in some cases, minimize an OF to achieve the best performance. To avoid routing loops, the RPL uses weights associated with the classification of each node in the graph. Such weights are considered in such a way that the child node classification is always higher than that of the parent node, increasing as it moves away from the root node.

#### 2.2.1. ETX Metric

The expected transmission count [8] is based on the expected number of data packet transmission attempts required to transmit the packet successfully. It is a link-based metric that emphasizes the rate of packet delivery between communication devices and not node-level statistics. This metric is expected to increase efficiency, as retransmission attempts are minimized by choosing a better link. However, Expected Transmission Count (ETX) does not consider the path length and can select longer paths. Also, channel conditions are dynamic, which is why frequent switching from parents is highly likely.

The ETX of a wireless link is the estimated average number of data and ACK frame transmissions required for the successful transmission of a packet [9]. As we know, the transfer of data can be in both directions. From the source to the destination, the node is defined as the forward direction, and the data transmission rate is $d_{f}$, which represents the probability of successful transmission in a positive direction. On the other hand, from the destination to the source node is defined as the reverse direction, and the data transmission rate is $d_{r}$. The ETX value is defined as follows:point-to-point communication: $ETX=\frac{1}{1-e_{pt}}$where the error probability ($e_{pt}$) is: $e_{pt}=1-d_{f}\times dr$and the bidirectional communication is: $ETX=\frac{1}{d_{f}\times d_{r}}$

#### 2.2.2. HOP Metric

For this metric, the preferred parent selection decision is based on the number of hops between the sending node and the collecting node. This simple metric only selects the path with a minimum number of hops. Clearly, this is a network-based metric that tries to emphasize the number of hops, rather than node or link statistics [8].

## 3. Related Work

Sensor nodes have limited battery power and can be anywhere, including areas with restricted/difficult access, making it hard to replace or recharge their batteries. Therefore, the energy consumption of each sensor node must be minimized to maximize the network lifetime. Many recent research works were proposed to investigate how to improve the wireless communication, e.g., in BLE [10,11], NB-IoT [12], and Wi-Fi [13].

Different performance evaluations show that RPL is an effective routing protocol because of its fast network reconfiguration, relatively short delays, and the ability of nodes to recover quickly from the loss of connectivity [14,15].

The problem of battery consumption in the RPL can be seen in Sharkawy et al. [16], where an objective function was proposed to maximize the node’s lifetime taking into account the so-called battery utility function. This OF is modeled as the remaining time interval, in seconds, that a node’s battery can contain. From its maximum value, the model calculates the time spent by a node in transmitting/receiving data and subtracts it from the last residual value of the battery.

Thomson et al. [17] showed the performance evaluation of the ETX, HOP, and RPL energy metrics in environments with extended transmission ranges varying the number of nodes, the topology, and the transmission/interference ranges. Some of the ideas in this article were based on the tests carried out as part of this study. Another one was conducted by Qasem et al. [18], which evaluated the performance of RPL in terms of two OFs, e.g., Hysteresis Objective Function (MRHOF) and OF0 in two different topologies (grid and random). To study the performance of OFs in the RPL, parameters such as PDR (Packet Delivery Ratio), energy consumption, and receive (RX) were considered.

However, the works conducted by Thomson et al. and Qasem et al. did not consider assessing energy consumption through network resilience, as presented by our research. In our work, the evaluation method was to obtain a percentage of the total number of nodes in the network to observe its behavior and assess the variation in energy consumption. The evaluation based on statistical methods shows that it is possible to find the best topology, the best metric, and the best objective function to be used by the RPL, paving the way for optimizing the restricted sensor network and its involved routing protocols.

## 4. Evaluation Proposal and Analysis

We investigated the impact of the network topology on the RPL, changing its metrics and associated objective functions. The evaluation was conducted to help the administrator choose the best position and configuration in an IoT environment based on WSN and the RPL algorithm. In another way, the results could be used to propose new RPL metrics that could use the virtual topology found to set the path routing, optimizing, and extending the network lifetime.

The metrics to build the RPLs’ DODAGswere selected, i.e., OF0 [19] and MRHOF [20]. Each scenario was run for 10 min, and each simulation was run ten times, applying a confidence interval of 95%. Finally, each scenario was executed with two different transmission and interference ranges: first, a transmission range of 50 m and an interference range of 100 m (a high interference with lower cover) and, second, a transmission range of 70 m and an interference range of 90 m (a scenario with higher interference and a longer range). The investigation was conducted using a complete factorial design using all possible combinations at all levels of all factors [21]. One investigation had *k* factors, where one factor *i* had $n_{i}$ levels and required *n* experiments, as follows: $n=\prod_{i=1}^{k}n_{i}$. There was a total of 1440 simulations, 480 for each topology.

### 4.1. Energy Consumption Evaluation

All experiments were conducted using the Cooja simulator from Instant Contiki 3.0 with real hardware configurations of Tmote Sky [22]. The evaluation is based on previous studies with a primary focus on the two OFs, OF0, and MRHOF. In Contiki, the use of OF0 resulted in the use of the hop count, which is the metric HOP, and the use of MRHOF based on the quality of links, which is the ETX metric; both were available in Cooja and were used.

### 4.2. Simulation Parameters

The simulations parameters can be seen in Table 1. Nine different scenarios were performed, with each scenario using a particular topology and number of nodes. These topologies used combinations of transmission range and interference. Scenarios 1, 4, and 7 were composed of one sink node and nine source nodes; Scenarios 2, 5, and 8 were composed of one sink node and 19 source nodes; while Scenarios 3, 6, and 9 were composed by one sink node and 29 source nodes; with all scenarios performed in 10 min. The three different numbers of nodes were chosen to evaluate the network scalability and its impact on energy consumption.

### 4.3. Results and Analysis

The results regarding energy consumption and node losses were compared in different scenarios with two metrics (ETX and HOP) and two transmission/interference ranges (50/100 m and 70/90 m). In all cases, the relation among the transmission/interference range from 50/100 m to 70/90 m and its significant effect on the RPL’s ability to create and maintain a DODAG instance was clear. This demonstrated that when the nodes were just within each other’s interference range, it caused great difficulty in exchanging messages to the degree that a DODAG could not be maintained. In the case of the ETX metric in the 50/100 m scenarios, it always had the highest energy consumption compared to the 70/90 m scenarios due to the smaller transmission area and the higher number of parents to change, so it needed to make more hops to reach the sink node. Furthermore, it always exchanged parents to maintain a good quality of the link.

Another critical point was the influence of disconnected nodes on network behavior. Disconnected nodes were nodes out of range, or offline (either due to a lack of battery or a duty-cycle strategy). Comments about the results will be presented during the text, and final remarks about it are presented in the last part of this section.

#### 4.3.1. Tree Topology

For each topology, three scenarios were investigated. For this, the first scenario was a 10 node tree topology with a 50/100 m and 70/90 m transmission/interference range (Figure 2. The second scenario was a tree topology with 20 nodes and a transmission/interference range of 50/100 m and 70/90 m (Figure 3). The third scenario was a tree topology with 30 nodes and a transmission/interference range of 50/100 m and 70/90 m (Figure 4).

The average energy consumption can be seen in Figure 5, Figure 6 and Figure 7 and in Table 2. Regarding the two metrics in use, the ETX metric with 50/100 m had the highest energy consumption in scenarios with disconnected nodes (without energy), consuming more than 25% of the total energy when the number of nodes disconnected was greater than 20%. Both metrics performed better in the 70/90 m transmission/interference range, with the lowest energy consumption in all scenarios where there were disconnected nodes for HOP with 70/90 m. As the percentile of disconnected nodes increased, both metrics showed slightly increased energy consumption, but the ETX metric was the most affected.

#### 4.3.2. Circular Topology

For the circular topology, we investigated three more scenarios: first, with ten nodes and a transmission/interference range of 50/100 m and 70/90 m (Figure 8, second, with 20 nodes and a transmission/interference range of 50/100 m and 70/90 m (Figure 9), and third, with 30 nodes and a transmission/interference range of 50/100 m and 70/90 m (Figure 10).

The energy consumption averages can be seen in Figure 11, Figure 12 and Figure 13 and in Table 3. Regarding the two metrics in use, the ETX metric with 50/100 m had the highest energy consumption in the scenarios with disconnected nodes, except in the case of 10 nodes where the highest consumption was for the ETX metric with 70/90 m. We believe that this occurred because the topology made it possible for almost all nodes to communicate directly with the sink node. At 70/90 m, it had greater consumption due to the use of transmission power. What could be observed concerning the metrics was that the lowest energy consumption was in all scenarios for HOP with disconnected nodes with 70/90 m. As the percentile of disconnected nodes increased, the two metrics slightly increased the energy consumption, but the ETX metric increased more in the scenarios with 20 and 30 nodes. This could be understood because of the nature of the ETX metric. Where different paths were possible from the source to the sink, it was necessary to perform an analysis and make a decision, increasing the energy consumption before starting the packets’ flow.

#### 4.3.3. Grid Topology

For this topology, we conducted three different scenarios: first, 10 nodes and a transmission/interference range of 50/100 m and 70/90 m (Figure 14), second, 20 nodes and a transmission/interference range of 50/100 m and 70/90 m (Figure 15), and third, 30 nodes with a transmission/interference range of 50/100 m and 70/90 m (Figure 16).

The mean energy consumption can be observed in Figure 17, Figure 18 and Figure 19 and Table 4. The ETX metric with 50/100 m had the highest energy consumption in scenarios with disconnected nodes. In the 10 node simulation case, there was little difference in the results compared to the 20 and 30 node simulations. Both metrics performed better in the transmission/interference range of 70/90 m, with the lowest energy consumption in all scenarios with nodes disconnected for HOP with 70/90 m. As the percentile of disconnected nodes increased, the two metrics slightly increased the energy consumption, but the ETX metric increased more, except for when there were ten nodes.

### 4.4. Discussion

As mentioned previously, to evaluate the different topologies, the comparison was applied using the paired observations method [21]. This process could help us to conclude which was the best metric in each scenario presented.

In the six simulated scenarios of the tree topology, the results showed that the HOP metric was equivalent in three scenarios (10 nodes with 50/100 m and 70/90 m, 20 nodes with 50/100 m) and better in three scenarios (20 nodes with 70/90 m, 30 nodes with 50/100 m and 70/90 m). In the six simulated scenarios of the circular topology, the results showed that the HOP metric was equivalent in two scenarios (10 nodes with 70/90 m, 20 nodes with 50/100 m) and better in four scenarios (10 nodes with 50/100 m, 20 nodes with 70/90 m, 30 nodes with 50/100 m and 70/90 m). In the six simulated scenarios of the grid topology, the results showed that the HOP metric was equivalent in two scenarios (10 nodes with 50/100 m, 20 nodes with 70/90 m) and better in four scenarios (10 nodes with 70/90 m, 20 nodes with 50/100 m, 30 nodes with 50/100 m and 70/90 m).

The benefits of knowing the topology of WSNs were clear, and it could help the administrator manage and design his/her network. Nevertheless, we know how hard it is to do this in the real world, where the nodes can be randomly arranged. However, if the topology could be considered for the administrator, as for the routing protocol, it could greatly help to increase the network lifetime. For example, a metric of the RPL could construct a virtual path considering the network topology observed. If the RPL used the distance perceived to build its DODAG, it could take advantage of the benefits shown in this work. Another relevant conclusion of our investigation was the correlation of the metrics ETX and HOP when we had more hops and possible routes. With the HOP metric, the most simple metric was applied, and the energy consumption was limited by it, considering only the route with fewer hops. However, the ETX metric was more sophisticated, considering not only the hops but also other parameters such as transmission packets, consequently consuming more processing to find the better path from the source to the destination. In this way, the metric HOP better fits a WSN with restrictions of the wireless signal coverage and energy consumption, besides behaving better in dense networks.

Thus, after comparing the topologies separately, the tree and circular topologies were evaluated. In this case, comparisons were made only with the HOP metric, which was the result of being the best metric in the assessments made by each separate topology. The results were from the six simulated scenarios, and the circular topology was better in all scenarios. Then, the circular topology was evaluated with the grid topology. Likewise, comparisons were made only with the HOP metric. The results were from six different scenarios, and the circular topology was better in all scenarios.

Regarding the three topologies evaluated, it could be concluded that a circular topology resulted in better performance concerning maintaining the number of nodes. The tree topology required considerably more of the first hop node and could also cause other bottlenecks in the network. The node density was critical in tree topology and needed to be carefully evaluated before being adopted in an IoT with RPL.

In Figure 20, it can be seen that in general, the topologies with the 50/100 transmission/interference range had higher energy consumption than the scenarios performed with the 70/90 m transmission/interference range. Furthermore, the circular topology had the lowest energy consumption, followed by the grid topology, and finally, the tree topology obtained the highest energy consumption. Therefore, it was clear that the circular topology was the best compared to the other topologies in the disconnected node scenarios with a high level of confidence.

## 5. Conclusions and Future Work

Efficient energy consumption is a challenging problem in battery-powered wireless sensor networks. The facts that each node has a limited battery charge and it is also impossible or impracticable to recharge the batteries, thus reducing energy consumption, are the keys to increasing the network’s lifetime.The scenario investigated was the PA for a coffee plantation in Southeast Brazil, and this research made clear the necessity to evaluate the benefits of the sensor’s location in the field. In this paper, the evaluation of the energy consumption of the RPL routing protocol was given for nine combinations of different scenarios in three distinct topologies, changing the number of nodes, the transmission/interference range, and the metric used. Based on the results obtained, we observed the best performance of the network’s energy consumption using a circular topology with the HOP metric. Our statistical results were achieved using the method of paired comparisons and indicated that for all topologies, the HOP metric obtained the lowest energy consumption compared to the ETX metric. When evaluated together, the circular topology had the lowest energy consumption compared to the other topologies with the HOP metric.

Regarding the three topologies evaluated, it could be concluded that a circular topology resulted in better performance concerning maintaining the number of nodes. The tree topology required considerably more of the first hop node, and could also cause other bottlenecks in the network. In conclusion, node density was significant in the tree topology and needed to be carefully evaluated before being adopted in an IoT with the RPL.

As future work, it is relevant to add knowledge of the RPL’s topology and metrics to compare with other existing routing protocols in WSNs, e.g., Lightweight On-demand Ad hoc Distance-vector Routing Protocol - Next Generation (LOADng) [23] and LOADng-IoT [24], and evaluate the RPL under other metrics and network topologies, as well as propose modifications to the RPL metrics to consider the context-awareness and its virtual topologies, and finally, examine authentication and authorization methods [25].

## Figures and Tables

**Figure 1 sensors-20-03814-f001:**
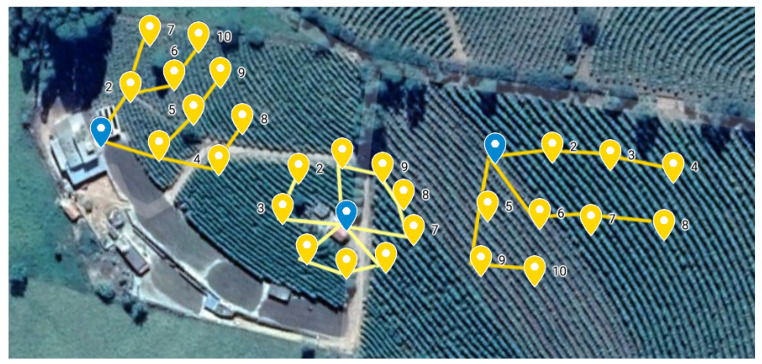
A real monitoring scenario for a coffee plantation with three different topologies, which are tree, circular, and grid topologies.

**Figure 2 sensors-20-03814-f002:**
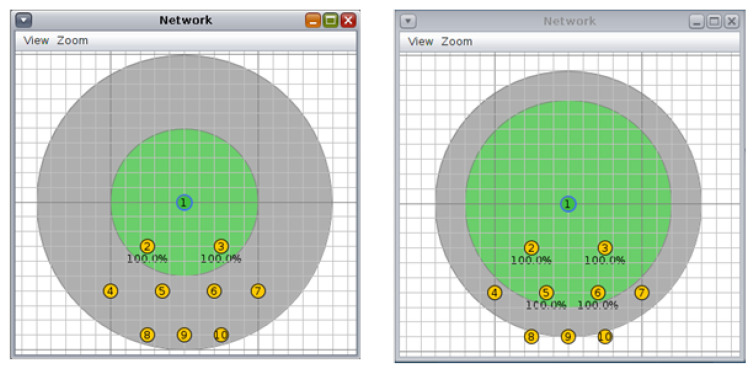
Scenario 1, 10 nodes, tree topology: (**left**) 50/100 m, (**right**) 70/90 m.

**Figure 3 sensors-20-03814-f003:**
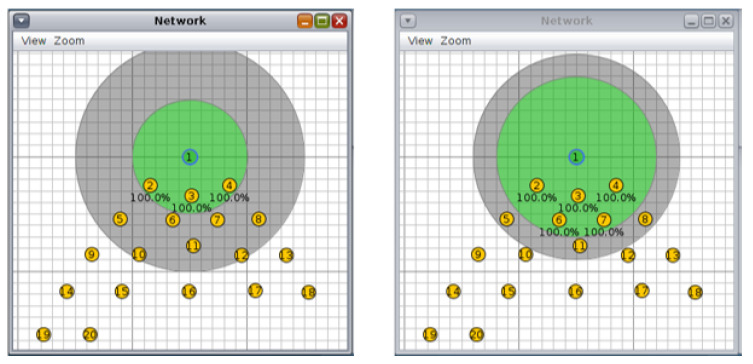
Scenario 2, 20 nodes, tree topology: (**left)** 50/100 m, (**right**) 70/90 m.

**Figure 4 sensors-20-03814-f004:**
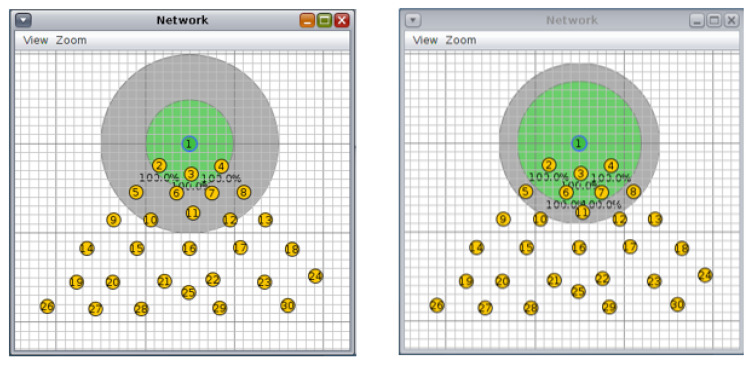
Scenario 3, 30 nodes, tree topology: (**left**) 50/100 m, (**right**) 70/90 m.

**Figure 5 sensors-20-03814-f005:**
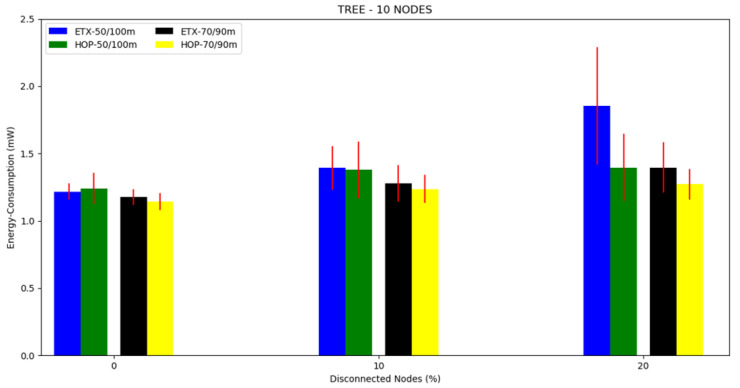
Scenario 1, 10 nodes: energy consumption versus disconnected nodes from the tree topology.

**Figure 6 sensors-20-03814-f006:**
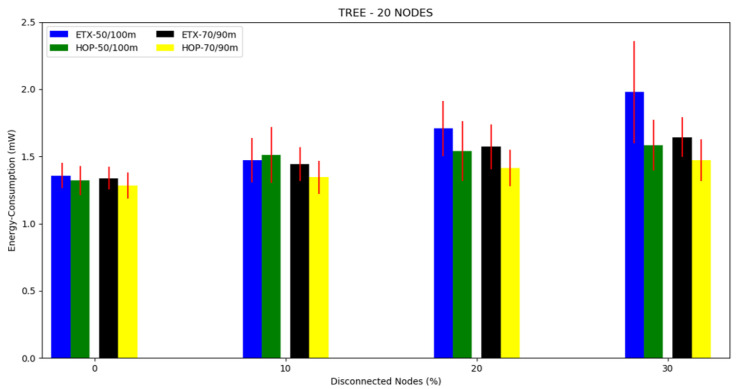
Scenario 2, 20 nodes: energy consumption versus disconnected nodes from the tree topology.

**Figure 7 sensors-20-03814-f007:**
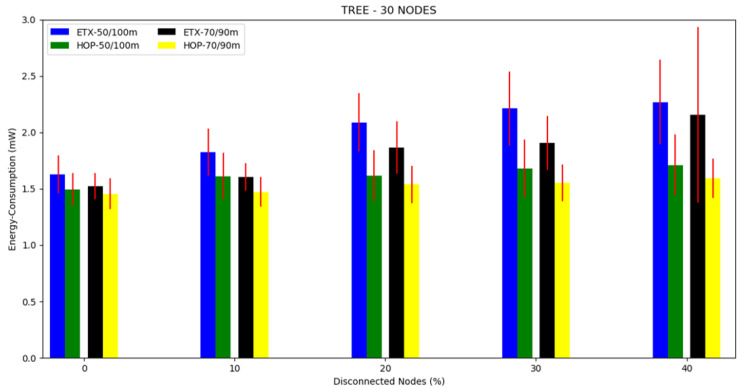
Scenario 3, 30 nodes: energy consumption versus disconnected nodes from the tree topology.

**Figure 8 sensors-20-03814-f008:**
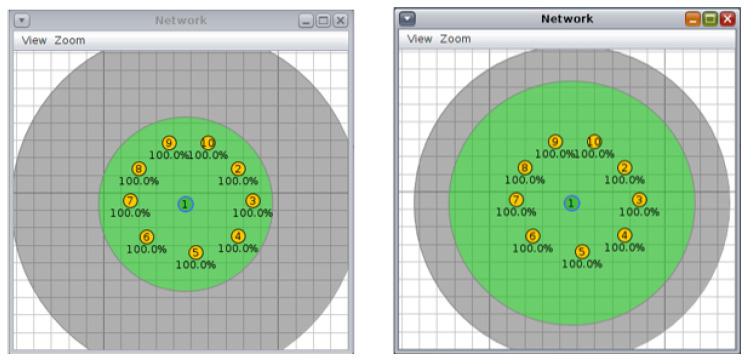
Scenario 4, 10 nodes, circular topology: (**left**) 50/100 m, (**right**) 70/90 m.

**Figure 9 sensors-20-03814-f009:**
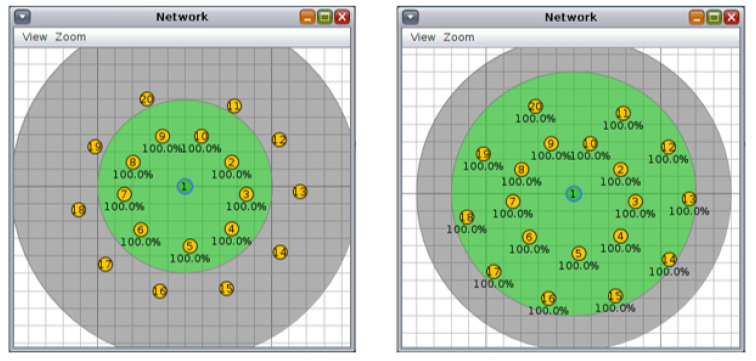
Scenario 5, 20 nodes, circular topology: (**left**) 50/100 m, (**right**) 70/90 m.

**Figure 10 sensors-20-03814-f010:**
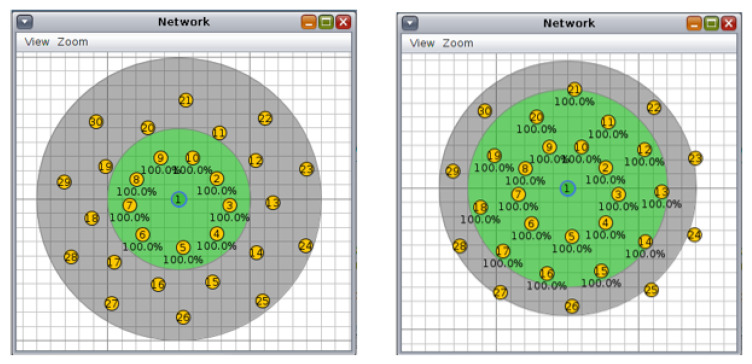
Scenario 6, 30 nodes, circular topology: (**left**) 50/100 m, (**right**) 70/90 m.

**Figure 11 sensors-20-03814-f011:**
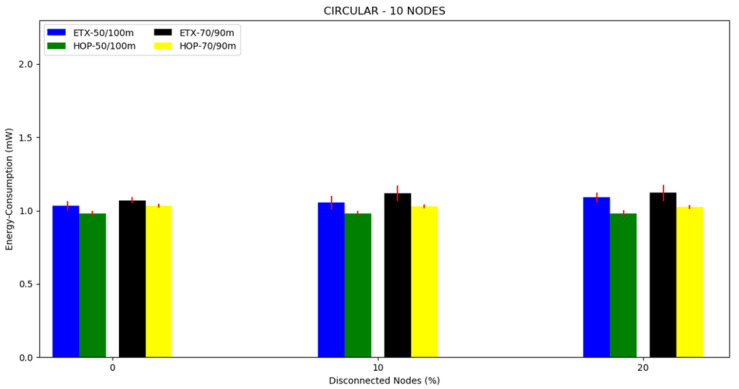
Scenario 4, 10 nodes: energy consumption versus disconnected nodes from the circular topology.

**Figure 12 sensors-20-03814-f012:**
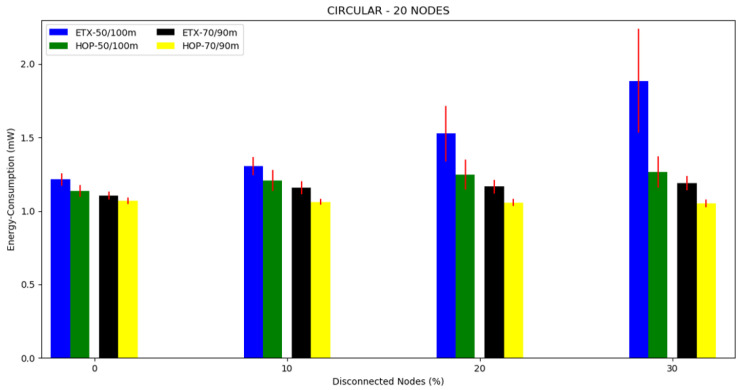
Scenario 5, 20 nodes, energy consumption versus disconnected nodes from the circular topology.

**Figure 13 sensors-20-03814-f013:**
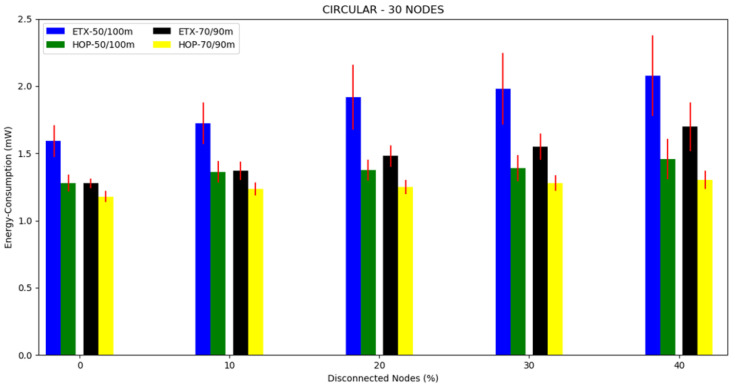
Scenario 6, 30 nodes: energy consumption versus disconnected nodes from the circular topology.

**Figure 14 sensors-20-03814-f014:**
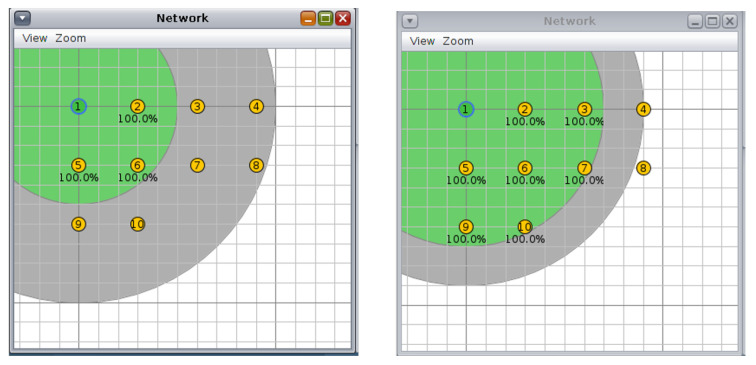
Scenario 7, 10 nodes: grid topology: (**left**) 50/100 m, (**right**) 70/90 m.

**Figure 15 sensors-20-03814-f015:**
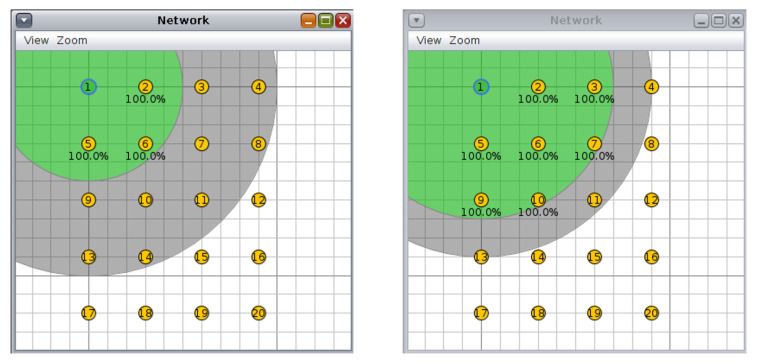
Scenario 8, 20 nodes, grid topology: (**left)** 50/100 m, (**right**) 70/90 m.

**Figure 16 sensors-20-03814-f016:**
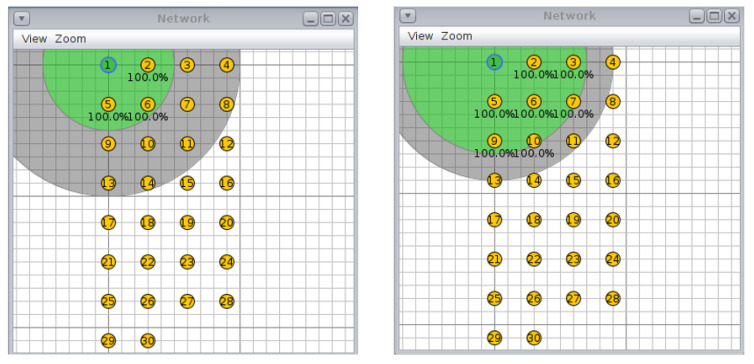
Scenario 9, 30 nodes, grid topology (**left**) 50/100 m, (**right**) 70/90 m.

**Figure 17 sensors-20-03814-f017:**
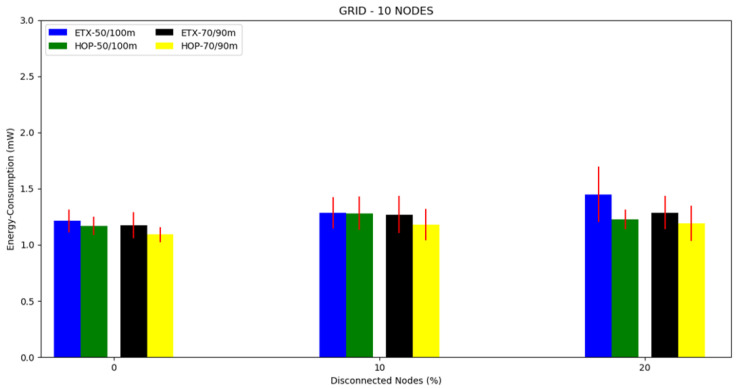
Scenario 7, 10 nodes: energy consumption versus disconnected nodes from the grid topology.

**Figure 18 sensors-20-03814-f018:**
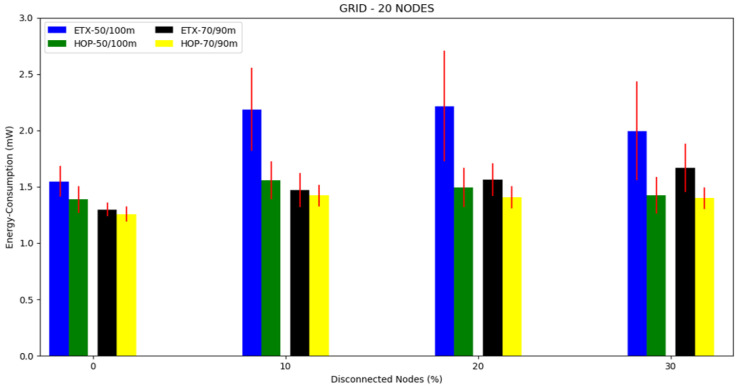
Scenario 8, 20 nodes: energy consumption versus disconnected nodes from the grid topology.

**Figure 19 sensors-20-03814-f019:**
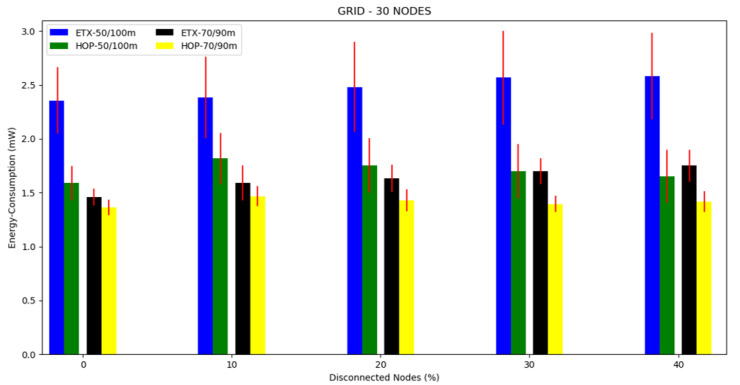
Scenario 8, 30 nodes: energy consumption versus disconnected nodes from the grid topology.

**Figure 20 sensors-20-03814-f020:**
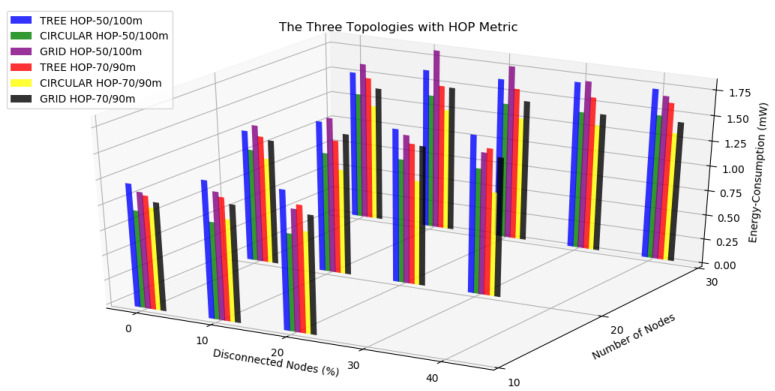
Energy consumption from the three topologies with the HOP metric.

**Table 1 sensors-20-03814-t001:** Simulation parameters. OF, Objective Function.

Parameters	Values	
Objective Functions	OF0	MRHOF
Metrics	HOP	ETX
Transmission/interference ranges	50/100 m, 70/90 m	
Topologies	Tree, circular, grid	
Simulation duration	10 min	
Number of nodes	10, 20, and 30	
Type of nodes	Sky Mote	
Wireless channel model	UDGM	

**Table 2 sensors-20-03814-t002:** Energy consumption of the tree topology.

			Energy Consumption (mW)			Energy Consumption (mW)	
Number of Nodes	Disconnected Nodes (%)	Transmission/ Interference	ETX	HOP	Transmission/ Interference	ETX	HOP
10	0	50/100 m	1.2173	1.2386	70/90 m	1.1774	1.1434
	10		1.3928	1.3777		1.2768	1.2367
	20		1.8560	1.3957		1.3965	1.2715
20	0	50/100 m	1.3576	1.3200	70/90 m	1.3386	1.2832
	10		1.4709	1.5115		1.4430	1.3438
	20		1.7069	1.5385		1.5716	1.4149
	30		1.9783	1.5827		1.6428	1.4723
30	0	50/100 m	1.6277	1.4936	70/90 m	1.5241	1.4550
	10		1.8235	1.6107		1.6044	1.4718
	20		2.0865	1.6153		1.8650	1.5381
	40		2.2123	1.6809		1.9048	1.5501
	50		2.2674	1.7096		2.1557	1.5930

**Table 3 sensors-20-03814-t003:** Energy consumption in the circular topology.

			Energy Consumption (mW)			Energy Consumption (mW)	
Number of Nodes	Disconnected Nodes (%)	Transmission/ Interference	ETX	HOP	Transmission/ Interference	ETX	HOP
10	0	50/100 m	1.0352	0.9807	70/90 m	1.0714	1.0322
	10		1.0542	0.9790		1.1173	1.0286
	20		1.0895	0.9792		1.1222	1.0230
20	0	50/100 m	1.2143	1.1361	70/90 m	1.1039	1.0688
	10		1.3056	1.2062		1.1576	1.0615
	20		1.5256	1.2476		1.1650	1.0568
	30		1.8854	1.2637		1.1884	1.0500
30	0	50/100 m	1.5911	1.2786	70/90 m	1.2773	1.1778
	10		1.7239	1.3621		1.3690	1.2342
	20		1.9176	1.3756		1.4777	1.2500
	40		1.9779	1.3902		1.5488	1.2792
	50		2.0759	1.4568		1.6978	1.3020

**Table 4 sensors-20-03814-t004:** Energy consumption in the grid topology.

			Energy Consumption (mW)			Energy Consumption (mW)	
Number of Nodes	Disconnected Nodes (%)	Transmission/ Interference	ETX	HOP	Transmission/ Interference	ETX	HOP
10	0	50/100 m	1.2124	1.1705	70/90 m	1.1744	1.0915
	10		1.2842	1.2812		1.2697	1.1806
	20		1.4484	1.2248		1.2868	1.1899
20	0	50/100 m	1.5482	1.3864	70/90 m	1.2977	1.2575
	10		2.1861	1.5578		1.4724	1.4210
	20		2.2158	1.4922		1.5615	1.4058
	30		1.9948	1.4259		1.6700	1.3994
30	0	50/100 m	2.3544	1.5900	70/90 m	1.4585	1.3630
	10		2.3837	1.8175		1.5898	1.4666
	20		2.4808	1.7541		1.6330	1.4277
	40		2.5662	1.7010		1.6993	1.3954
	50		2.5825	1.6530		1.7517	1.4167

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
