# Peer review of "Energy Consumption Evaluation of a Routing Protocol for Low-Power and Lossy Networks in Mesh Scenarios for Precision Agriculture"

_sensors, 2020, doi:10.3390/s20143814_

Round 1

Reviewer 1 Report

The authors provide an evaluation for a routing protocol called RPL. A lot of work have been done by the authors. The presentation of the manuscript is good.

However, the reviewer is thinking that why the authors only focus on only one specific protocol as RPL. How about the others. The reviewer thinks that many other protocols deserve to be compared with.

In addition, this manuscript sounds like a review or an evaluation, not an article. Please provide a novel idea if applicable.

Author Response

Dear Reviewer:

The authors would like to thank you for the comments and suggestions to improve our paper. In this follow, we have written this letter answering each one question.

All modifications to the original manuscript were highlighted in the new paper submitted.

The article was updated by introducing a real IoT scenario, considering the benefits of using mesh communication to monitor it. The scenario is a precision agriculture (PA) field, a coffee plantation in Brazil. Our investigation and results can help the network manager to plan his WSN. 

Reviewer #1

Q: The authors provide an evaluation for a routing protocol called RPL. A lot of work have been done by the authors. The presentation of the manuscript is good.

A: We were glad to hear it. Thank you.

Q: However, the reviewer is thinking that why the authors only focus on only one specific protocol as RPL. How about the others. The reviewer thinks that many other protocols deserve to be compared with.

A: We improve the text highlighting why RPL was investigated and not another routing protocol as LOADng. Our justification is about the RPL is the well-known protocol and standardized by IETF in RFC, and we are focusing on topology network impact, more than the LLN protocol. The authors consider evaluating the impact of other LLN protocols and other metrics in future work.

Q: In addition, this manuscript sounds like a review or an evaluation, not an article. Please provide a novel idea if applicable.

A: The authors review all manuscript content, providing a fluid reading about the benefits of using a better topology for PA and IoT.

Reviewer 2 Report

This is to study energy consumption evaluation on routing protocol for low and lossy networks in mesh scenarios. It looks interested issue, however because of next reasons, I decided major revision to this paper.

  1. It is not clear the mesh scenario. This paper’s target is a mesh scenario, then, the mesh scenarios should be clearer.
  2. There are no existing problem in abstract. To define the existing problem is important, because the readers could realize why the authors decided to write this research.
    1. Also, hard to detect the idea to solve. (On the other hand, how much improve by writing 15%, and 30% is so good.)
  3. How close the material in Background section to the suggestion in this paper? The material that the authors wrote feels some broad. Write more specific.
  4. In section 3.3.3 Grid topology, missed closed ‘)’ after (Figure 15.
  5. Overall there are three categorized such as 10 nodes, 20 nodes, and 30 nodes. But hard to understand what differences them. Write more remarkable what strong is.

Author Response

Dear Reviewer:

The authors would like to thank you for the comments and suggestions to improve our paper. In this follow, we have written this letter answering each one question.

All modifications to the original manuscript were highlighted in the new paper submitted.

The article was updated by introducing a real IoT scenario, considering the benefits of using mesh communication to monitor it. The scenario is a precision agriculture (PA) field, a coffee plantation in Brazil. Our investigation and results can help the network manager to plan his WSN. 

Reviewer #2

Q: This is to study energy consumption evaluation on routing protocol for low and lossy networks in mesh scenarios. It looks interested issue, however because of next reasons, I decided major revision to this paper.

A: The authors thank you for your observations.

Q: It is not clear the mesh scenario. This paper’s target is a mesh scenario, then, the mesh scenarios should be clearer.

A: The paper was review considering it. All scenarios are now explicit to explain which mesh networks were evaluated.

Q: There are no existing problem in abstract. To define the existing problem is important, because the readers could realize why the authors decided to write this research. Also, hard to detect the idea to solve. (On the other hand, how much improve by writing 15%, and 30% is so good.)

A: The abstract was modified following your comments. Now it is more clear the scenario considered by us to conduct this research. Thank you.

Q: How close the material in Background section to the suggestion in this paper? The material that the authors wrote feels some broad. Write more specific.

A: We conducted a review in this section to help the reader to understand the main concepts.

Q: In section 3.3.3 Grid topology, missed closed ‘)’ after (Figure 15.

A: Thank you. We fixed it.

Q: Overall there are three categorized such as 10 nodes, 20 nodes, and 30 nodes. But hard to understand what differences them. Write more remarkable what strong is.

A: A new paragraph explaining the scenario and scalability evaluation was written to support our chose by the number of nodes.

Reviewer 3 Report

The paper investigates the impact of the network topology on the energy consumption in the RPL routing protocol.

The subject of the paper is worthy of investigation, and the paper is well written. I quite liked reading it. However, it requires some revisions and additional experiments before being accepted for publication. Here are my comments:

The Related work section should be after the Introduction.

  1. Section 2 (Related work) is missing some recent state-of-the-art works related to this research. Please update your paper and comparison. For example: 
  • Nikodem, M.; Bawiec, M. Experimental Evaluation of Advertisement-Based Bluetooth Low Energy Communication. Sensors 2019, 20, 107.
  • Bulić, P.; Kojek, G.; Biasizzo, A. Data Transmission Efficiency in Bluetooth Low Energy Versions. Sensors 201919, 3746.
  • Basu, S.S.; Haxhibeqiri, J.; Baert, M.; Moons, B.; Karaagac, A.; Crombez, P.; Camerlynck, P.; Hoebeke, J. An End-To-End LwM2M-Based Communication Architecture for Multimodal NB-IoT/BLE Devices. Sensors 2020, 20, 2239. 

The results section is weak. The authors do not provide information on how many measurements are performed nor standard deviations. The measurement setup should be described in detail. The charts should contain error bars (i.e., confidence interval or std deviations).

Author Response

Dear Reviewer:

The authors would like to thank you for the comments and suggestions to improve our paper. In this follow, we have written this letter answering each one question.

All modifications to the original manuscript were highlighted in the new paper submitted.

The article was updated by introducing a real IoT scenario, considering the benefits of using mesh communication to monitor it. The scenario is a precision agriculture (PA) field, a coffee plantation in Brazil. Our investigation and results can help the network manager to plan his WSN. 

Reviewer #3

Q: The paper investigates the impact of the network topology on the energy consumption in the RPL routing protocol. The subject of the paper is worthy of investigation, and the paper is well written. I quite liked reading it. However, it requires some revisions and additional experiments before being accepted for publication. Here are my comments:

A: Thank you. We are happy to hear this.

Q: The Related work section should be after the Introduction.

A: This section was moved after the Introduction.

Q: Section 2 (Related work) is missing some recent state-of-the-art works related to this research. Please update your paper and comparison. For example: 

  1. Nikodem, M.; Bawiec, M. Experimental Evaluation of Advertisement-Based Bluetooth Low Energy Communication. Sensors 2019, 20, 107.
  2. Bulić, P.; Kojek, G.; Biasizzo, A. Data Transmission Efficiency in Bluetooth Low Energy Versions. Sensors 2019, 19, 3746.
  3. Basu, S.S.; Haxhibeqiri, J.; Baert, M.; Moons, B.; Karaagac, A.; Crombez, P.; Camerlynck, P.; Hoebeke, J. An End-To-End LwM2M-Based Communication Architecture for Multimodal NB-IoT/BLE Devices. Sensors 2020, 20, 2239. 

 A: The papers 1 and 3 were cited now by us. Another paper from Sensors was added too.

Q: The results section is weak. The authors do not provide information on how many measurements are performed nor standard deviations. The measurement setup should be described in detail. The charts should contain error bars (i.e., confidence interval or std deviations).

A: All of the results were refined, and the charts and statistical methods were adopted to validate it. Now we have a new paragraph at the beginning of the results explaining this. The bar charts were replotted with error bars.

Round 2

Reviewer 1 Report

The manuscript does not have abstract.

Author Response

Thank you so much.

The abstract was added and the last review finalized.

The authors appreciate your help.

Reviewer 2 Report

I am so pleased to accept this paper to be published, I am sure this paper has updated following 1st round review.

But, I can't see the abstract section. So, I would like the authors to add the the abstract section in  the final version.

Author Response

The authors appreciate your help. Thank you so much.

The abstract was insert and the last review finalized.